# High Mobility Group Box 1 (HMGB1): Potential Target in Sepsis-Associated Encephalopathy

**DOI:** 10.3390/cells12071088

**Published:** 2023-04-04

**Authors:** Bram DeWulf, Laurens Minsart, Franck Verdonk, Véronique Kruys, Michael Piagnerelli, Mervyn Maze, Sarah Saxena

**Affiliations:** 1Department of Anesthesia—Critical Care, AZ Sint-Jan Brugge Oostende AV, 8000 Bruges, Belgium; 2Department of Anesthesia, Antwerp University Hospital (UZA), 2650 Edegem, Belgium; 3Department of Anesthesiology and Intensive Care, GRC 29, DMU DREAM, Hôpital Saint-Antoine and Sorbonne University, Assistance Publique-Hôpitaux de Paris, 75012 Paris, France; 4Laboratory of Molecular Biology of the Gene, Department of Molecular Biology, Free University of Brussels (ULB), 6041 Gosselies, Belgium; 5Department of Intensive Care, CHU-Charleroi, Université Libre de Bruxelles, 6042 Charleroi, Belgium; 6Experimental Medicine Laboratory (ULB Unit 222), CHU-Charleroi, Université Libre de Bruxelles, 6110 Montigny-le-Tilleul, Belgium; 7Center for Cerebrovascular Research, Department of Anesthesia and Perioperative Care, University of California San Francisco, San Francisco, CA 94143, USA

**Keywords:** sepsis, sepsis-associated encephalopathy, HMGB1, high mobility group box 1

## Abstract

Sepsis-associated encephalopathy (SAE) remains a challenge for intensivists that is exacerbated by lack of an effective diagnostic tool and an unambiguous definition to properly identify SAE patients. Risk factors for SAE development include age, genetic factors as well as pre-existing neuropsychiatric conditions. Sepsis due to certain infection sites/origins might be more prone to encephalopathy development than other cases. Currently, ICU management of SAE is mainly based on non-pharmacological support. Pre-clinical studies have described the role of the alarmin high mobility group box 1 (HMGB1) in the complex pathogenesis of SAE. Although there are limited data available about the role of HMGB1 in neuroinflammation following sepsis, it has been implicated in other neurologic disorders, where its translocation from the nucleus to the extracellular space has been found to trigger neuroinflammatory reactions and disrupt the blood–brain barrier. Negating the inflammatory cascade, by targeting HMGB1, may be a strategy to complement non-pharmacologic interventions directed against encephalopathy. This review describes inflammatory cascades implicating HMGB1 and strategies for its use to mitigate sepsis-induced encephalopathy.

## 1. High Mobility Group Box 1 (HMGB1)

HMGB1 is an abundant nuclear protein that is highly conserved in mammals and belongs to the category of high molecular group (HMG) proteins that were first identified as chromatin-associated proteins in calf thymus and were so designated because of their high electrophoretic mobility [1]. The HMG superfamily is subdivided into three subgroups, namely HMGA, HMGB and HMGN, based on structural features; the HMGB protein subgroup includes three members, HMGB1, 2 and 3. HMGB1 is the most studied member of the HMG superfamily of proteins and the lethality of HMGB1 deficiency in mice shortly after birth highlights its essential role in survival [2]. Each of the HMGB1–3 proteins is composed of two positively charged DNA-binding domains called Box A and Box B and a negatively charged C-terminal acidic domain [3]. In the nucleus, HMGB1 is involved in multiple important processes such as DNA replication and repair, chromatin remodeling, transcription and genome stabilization [4,5]. HMGB1 binds DNA non-specifically through the L-shaped Box A and Box B interaction with the minor groove of B-type DNA [6]. The acidic C-terminal tail plays an important role as it regulates HMGB1 DNA-and chromatin-binding ability by shielding Boxes A and B from other interactions (Figure 1) [7,8].

HMGB1 contains three cysteine (“C”) residues at positions 23, 45 and 106 which can be oxidized. C23 and C45, located in Box A, can form a disulfide bridge and in strong oxidative conditions, all three cysteines can be fully oxidized and form sulfonyl groups. The redox state of these cysteine residues strongly influences HMGB1 functional activities as referred to later [9]. Two nuclear localization signals (NLSs) present in Boxes A and B mediate HMGB1 import into the nucleus by the importin-alpha/importin-beta nuclear transport receptor. The nuclear translocation of HMGB1 is strongly regulated by post-translational modifications within the NLS. Acetylation, phosphorylation and methylation of the amino acid residues of lysine and serine within the NLS neutralize the global positive charge and impair HMGB1’s ability to interact with nuclear carriers, thereby strongly reducing its nuclear accumulation. Additionally, acetylation of lysines in NLS promotes HMGB1 translocation from the nucleus to the cytoplasm and was shown to be controlled by histone deacetylases and type 1 and type 2 IFNs/JAK/STAT1 signaling pathways [10,11,12]. Phosphorylation of serines in the NLS can be induced by pro-inflammatory cytokines in immune cells and favors HMGB1 cytoplasmic accumulation [13]. Methylation of lysine 42 in NLS1 weakens DNA-binding activity of HMGB1, facilitating passive diffusion of HMGB1 out of the nucleus [14]. Translocation of HMGB1 from the nucleus to the cytoplasm, occurring upon immune cell activation or injury, promotes autophagy by binding to the autophagy mediator beclin-1 [15]. Cytoplasmic accumulation of HMGB1 activates the inflammasome and promotes pyroptosis, leading to the release of several danger-associated molecular patterns (DAMPs) including HMGB1 itself [16].

HMGB1 can be passively released through cell damage and is also secreted by exocytosis through secretory lysosomes, exosomes and microparticles from activated macrophages. As mentioned earlier, HMGB1 is released upon cell death by pyroptosis. Notably, the passively released HMGB1 originates from the nucleus and is not acetylated and fully reduced. In contrast, the secreted and pyroptosis-associated form of HMGB1 originates from the cytoplasm, is acetylated and contains a C23–C45 disulfide bond [17]. These HMGB1 isoforms have different signaling activities as described later.

HMGB1 was first described as a secreted product of LPS-activated monocytes/macrophages. HMGB1 is released into the extracellular compartment in response to various stressors, pathogen-associated molecular patterns (PAMPs) and pro-inflammatory cytokines and is referred to as an endogenous alarmin [18,19]. Extracellular HMGB1 can bind up to 14 different cell receptors depending on its redox state [9]. Moreover, binding to several receptors implies the association of HMGB1 with interacting partners, often leading to a synergistic interaction of HMGB1 in post-receptor signaling. Thus far, three receptors have been well characterized to bind the different HMGB1 isoforms [20]. The first one is CXCR4 which binds fully reduced HMGB1 released upon tissue damage in association with the chemokine CXCL12 and initiates chemotaxis in a synergistic manner compared to CXCL12 alone [21]. The TLR4-MD2 receptor and receptor for advanced glycation end products (RAGE) have been extensively characterized as specific HMGB1 receptors [20,22]. HMGB1 activates the TLR4-MD2 LPS receptor through its association with TLR4 co-receptor MD2 and triggers signaling cascades leading to the production of pro-inflammatory cytokines in monocytes/macrophages [23]. Detailed characterization of HMGB1 binding to the TLR4-MD2 receptor revealed that only the disulfide-containing HMGB1 isoform binds to MD2 [23]. Moreover, the development of a tetrapeptide impairing HMGB1 binding to MD2 without affecting LPS–MD2 interaction indicates that HMGB1 and LPS bind to distinct epitopes of MD2 which accounts for the synergistic interaction of LPS and HMGB1 in triggering inflammation [23].

The RAGE is expressed at the surface of many cell types. In contrast to TLR4-MD2, the RAGE can be bound by all HMGB1 redox forms including the fully oxidized C106 sulfonyl form of HMGB1. The capacity of the RAGE to bind fully reduced HMGB1 accounts for the inflammatory response induced by tissue damage triggering further passive release of reduced nuclear HMGB1. Recently, it was shown that fully reduced HMGB1 promotes LPS internalization by inflammatory macrophages in a RAGE-dependent manner. [24] Cytosolic LPS activates caspase-11 and gasdermin D-mediated pyroptosis, contributing to further DAMP release and inflammation upon sepsis [24].

Binding of HMGB1 to the RAGE triggers HMGB1 endocytosis directing HMGB1 to lysosomes, leading to its destruction and release of its lysosomic content into the cytosol. As HMGB1 co-binds to the RAGE in association with diverse ligands such as LPS, DNA, RNA, nucleosomes, histones and IL-1, it is thought that its release into the cytosolic compartment contributes to inflammatory signaling by activating endosomal and cytosolic “danger” receptors. Therefore, besides acting alone as a potent mediator of inflammation, HMGB1 acts in synergy with various DAMPs and PAMPs to trigger the inflammatory response [20].

Two distinct RAGE-binding epitopes in HMGB1 have been identified. The first one was mapped between amino acids 150 and 183 of the HMGB1 sequence and was shown to mediate neurite outgrowth in fetal mouse brain without signs of inflammation [25]; the second epitope was identified in Box A (23–50) and shown to reverse apoptosis-induced tolerance in dendritic cells [26].

The pro-inflammatory activities of HMGB1 are neutralized by several mechanisms contributing to the resolution phase of inflammation. One of the neutralization processes relies on HMGB1 binding by haptoglobin and subsequent CD163-mediated uptake by circulating macrophages. Internalization of HMGB1–haptoglobin complexes by macrophages induces the expression of anti-inflammatory proteins such as heme oxygenase-1 and IL-10, protecting against sepsis-induced lethality [27]. Moreover, HMGB1 can be trapped by the complement component C1q, leading to the assembly of a multimolecular signaling complex with the RAGE and leukocyte-associated Ig-like receptor 1 (LAIR-1) that promotes macrophage polarization to the M2-like anti-inflammatory phenotype [28]. The HMGB1–RAGE axis also controls the adaptative immune response by favoring the migration and survival of regulatory T cells which correspond to a subset of T lymphocytes downregulating T cell responses and maintaining immunological tolerance [29]. Furthermore, the soluble CD52 released upon T cell activation binds to HMGB1 and favors CD52 ligation to the Siglec-10 receptor, leading to the inactivation of T cell receptor signaling (Figure 2) [30].

HMGB1 is a major alarmin that disrupts homeostasis by triggering complex signaling, processes leading to inflammation. Therefore, in pathological contexts such as sepsis, HMGB1 constitutes a potential target for clinical interventions.

## 2. Sepsis-Associated Encephalopathy

Sepsis is a life-threatening organ dysfunction caused by a dysregulated host response to a bacterial, viral or fungal infection and occurs in nearly ~50 million patients globally; it is the main of cause of death in intensive care unit (ICU) patients with 11 million deaths worldwide in 2017 [31]. The presence of an organ dysfunction (cardiovascular, renal, coagulation, pulmonary, liver and brain) in septic patients increases the in-hospital mortality to >10% [32]. Sepsis-associated encephalopathy (SAE) is defined as a diffuse cerebral dysfunction accompanying sepsis, without central nervous system infection, structural abnormality or any other cause of encephalopathy [33,34].

SAE is thought to be the most common cause of encephalopathy in the medical–surgical ICU [35]. SAE manifests itself as a spectrum of disturbed cerebral function from mild delirium, through confusion, to coma. Due to this wide range of clinical presentations, however, incidence remains difficult to define [36,37,38]. Based on 2513 septic patients in 12 French ICUs (from 1997–2014), Sonneville et al. reported an SAE incidence of 53% derived from Coma Glasgow Scale < 15, together with abnormal neuropsychological findings consistent with delirium (i.e., inattention, disorientation, altered thinking, psychomotor slowing and/or agitation) [37]. SAE is also associated with a higher risk of developing long-term cognitive impairment in survivors of sepsis and a higher susceptibility to developing neuropsychiatric diseases such as dementia, depression, anxiety and post-traumatic stress disorder [38,39,40].

In the BRAIN-ICU study, which included 821 mechanically ventilated ICU patients for respiratory failure or for shock, Pandharipande et al. reported that at 3 months after ICU stay, 40% of patients had global cognitive scores below the general population and 26% even had similar scores to patients with mild Alzheimer’s disease [41]; this cognitive decline persisted at 12 months post-ICU stay.

## 3. Pathogenesis of SAE

The underlying mechanisms of SAE are complex and involve neurological structures responsible for cognitive domains such as memory, attention and consciousness. The pathogenic processes involved in SAE remain poorly understood and may be due to neuroinflammation and oxidative stress [42] and involve the activation of microglial cells [43], dysfunction of neurovascular coupling [44] and metabolic derangements [45].

Within the brain, microglial cells contribute to both innate and adaptive immune responses and are involved in the maintenance of brain homeostasis [46]; its surface receptors interact with the peripheral immune system through a variety of mechanisms, primarily by binding to cytokines, but also recognizing molecules associated with damage and pathogens, known as DAMPs and PAMPs, through their Toll-like receptors (TLRs) [47]. TLRs are activated at early stages by systemic inflammation in the setting of sepsis [48] and can lead to persistent neuroinflammation, which is described up to 14 days after sepsis induction in rodent models [49]. Microglial activation is characterized by morphological, functional and metabolic changes that can alter the phenotype, ranging from pro-inflammatory (M1), anti-inflammatory (M2a), to immunomodulatory (M2b) [50]. The neurotoxic consequences are usually considered secondary to pro-inflammatory phenotypes whereas anti-inflammatory phenotypes could be neuroprotective [51]. Recently, a unique catalog of phenotype markers was devised from rigorous observations of the effects of polarizing stimuli (LPS and IL-4) in primary culture of microglia; the authors recommend that these markers, rather than the M1–M2 characterization, be used in arbitrating the phenotype of microglia [52]. Cytokines, nitric oxide, excitatory neurotransmitters and metabolites (such as reactive oxygen species) mediate the microglia-mediated neurotoxicity through an increase in neuronal excitability and are consistent neuropathologic features of SAE in both animal models and patients. Interestingly, at early stages, the manner in which microglial cells respond to sepsis varies depending on the specific area of the brain [53], and the hippocampus is particularly affected by sepsis [54]. While research on both animals and humans during sepsis has consistently revealed microglial activation, even in the absence of known brain infections, recent findings have challenged this understanding, as evidence of both bacterial genetic material and live bacteria has been found in animals and humans without resulting in infectious inflammation of the brain [55].

Multiple experimental interventions have shown promising results in modulating microglial activity, including minocycline [56], ketamine [57], cholinergic inhibition [58] and vagal nerve stimulation [59] as well as blocking potassium channels needed for its activation [60]; regarding the latter, pre-clinical data showed that pre-operative blocking of Kv1.3 prevented peri-operative cognitive decline, reduced hippocampal IL-6 levels and prevented microglial activation [60]. Some studies have shown that tight glucose control can dampen microglial activation [61]. However, a randomized clinical trial that aimed to decrease microglial activation in a critical care setting by restoring cholinergic inhibition failed, increasing mortality, suggesting that manipulating microglia can be hazardous [62].

In addition to the direct effects of microglial activation on neurons, neuroinflammation may also contribute to SAE through its effects on the microvasculature of the brain. Inflammation can lead to the formation of microthrombi in the brain, which can cause ischemia and further neuronal injury [63]. Additionally, the activation of the complement cascade can lead to the formation of the membrane attack complex [64], which can damage the endothelial cells, leading to blood–brain barrier breakdown and increased permeability (Figure 3).

Other cells involved in neuroinflammation are astrocytes, specialized glial cells that form networks that provide energy and growth factors to neurons while also maintaining the homeostasis of the neuronal microenvironment. Astrocytes also regulate synaptic activity by incorporating glutamate and releasing neurotransmitters [65]. Astrocytes interact with endothelial cells and play a role in the formation and integrity of the blood–brain barrier. Studies in animals have shown astrocyte dysfunction or proliferation during SAE, but neuropathologic studies in humans have not found similar results [33]. The lack of astrocytic alterations in humans with SAE contrasts with evidence of blood–brain barrier impairment [66]. However, most studies have only relied on immunostaining of the glial fibrillary acid protein and morphologic analysis, which are insufficient to detect molecular changes or investigate the astroglial network or neuroglial interactions.

Eventually, neuroinflammation increases the metabolic and bioenergetic demand, which leads to oxidative stress and mitochondrial dysfunction. Early mitochondrial dysfunction has been detected in the brains of septic animals [45], resulting in the production of reactive oxygen or nitrogen species, affecting both glial cells and neurons [67]. The experimental benefit associated with tight blood glucose control might be a result of reduced mitochondrial dysfunction and oxidative stress.

## 4. Role of HMGB1 in SAE Development

HMGB1 has been identified as a key player in the inflammatory response during sepsis. It is released by immune cells and necrotic tissue and acts as a powerful neuroinflammatory mediator [68]. When HMGB1 is present in the extracellular environment, it interacts with macrophage/monocyte receptors such as TLR4 [69] and the RAGE [70], resulting in increased chemotaxis and activation of the NF-κB signaling pathway. This activates gene expression and the release of pro-inflammatory cytokines, including IL-6 and IL-1β. Strategies that target HMGB1 through the use of molecular inhibitors [71] or by inhibiting HMGB1 signaling pathways [72] have been shown to be effective in reducing mortality in sepsis models and altering early cytokine profiles to patterns observed in survivors.

Although there are limited data available about the role of HMGB1 in neuroinflammation following sepsis, it has been implicated in other neurologic disorders with neuroinflammatory features such as traumatic brain injury (TBI) [73] and peri-operative neurocognitive disorders (PNDs) [66]. In the brain, HMGB1 is expressed in all cell types, including microglia. Overexpression of extracellular HMGB1 has been observed in several neuroinflammatory conditions, including TBI [73], subarachnoid hemorrhage (SAH) [74] and neurodegenerative diseases [75]. The active translocation of HMGB1 from the nucleus to the extracellular space has been found to trigger neuroinflammatory reactions and damage the blood–brain barrier. Recently, studies have demonstrated that HMGB1 was upregulated in microglia in neonatal hypoxic–ischemic (HI) encephalopathy [71]. Furthermore, HMGB1 is able to regulate the M1/M2 phenotypic polarization of microglia, leading to cortical injury [76] and also to changes in the morphology and function of endothelial cells and pericytes in an experimental model of the blood–brain barrier [77]. Therefore, HMGB1 has emerged as an extracellular target against a diverse range of CNS disorders, potentially including SAE. Targeted therapy against HMGB1 has been shown to confer neuroprotective effects against these CNS disorders mainly by inhibiting its translocation and downregulating the immune response [73,75].

## 5. Neutralizing HMGB1

HMGB1 can be neutralized or blocked by a variety of methods, including the use of anti-HMGB1 antibodies or synthesized antagonists [78]. Anti-HMGB1 antibodies are a type of immunotherapy that specifically targets and neutralizes HMGB1. Anti-HMGB1 monoclonal antibodies, the pharmacological inhibitor glycyrrhizin which binds directly to both HMG boxes, inhibiting its chemoattractant and mitogenic properties, or HMGB1 interference (shRNA) have been shown to be effective in inhibiting the neuroinflammatory response post-TBI [73,79], in SAH [80,81] or in neurodegenerative-like experimental settings [82]. Experimental studies highlighted improved functional recovery, reduced lesion volume and reduced activation of the M1 phenotype and enhanced activation of the M2 phenotype of microglia/macrophages.

Another HMGB1 neutralization process is through the use of synthesized small molecules that can prevent HMGB1 from activating the immune response in the central nervous system (CNS). These antagonists can either block the release of HMGB1 from cells or bind to the receptors for HMGB1 on microglial cells, acting as a decoy. Several HMGB1 antagonists are currently being tested in pre-clinical and clinical trials for treating various neuroinflammatory conditions [83,84]. For instance, ethyl pyruvate, an aliphatic ester derived from the endogenous metabolite pyruvic acid, that can neutralize HMGB1, has been found to have biological effects in several diseases [85]. Omega-3 polyunsaturated fatty acids (ω-3 PUFAs) also possess a neuroprotective effect, which is attributed to the modulation of inflammatory pathways, particularly of the HMGB1 pathway [86]. After TBI, rats that received PUFA exhibited a decrease in microglial activation and decreased levels of IL-1β, IL-6 and HMGB1 [87]. Several other molecules also demonstrated beneficial effects in experimental subarachnoid hemorrhage (SAH), including resveratrol, melatonin and AG490 (inhibitor of JAK/STAT3); these agents mainly inhibit HMGB1 expression and release, further ameliorating neural apoptosis and brain edema and downregulating SAH-induced inflammatory markers [87].

However, a recent study has shown that the effects of deleting the protein HMGB1 in an experimental traumatic brain injury model can have negative effects. The authors found that deleting HMGB1 did not prevent brain swelling or protect the blood–brain barrier and did not improve survival of brain cells or reduce tissue loss in comparison to normal mice [88].

These findings indicate that more research is needed to fully understand the risks/benefits of HMGB1 in traumatic brain injury and to carefully select the best methods to block HMGB1 for neurotherapeutic indications.

## 6. Possible Role of HMGB1 in the Risk Factors for SAE Development

Several studies suggest that increasing age might be an independent risk factor for severe sepsis and SAE development (see Table 1) [37,89,90,91,92]. Roughly 60% of ~200,000 septic patients were older than 65 years with 80% case fatality [93].

Plasma levels of HMGB1 tend to also increase with age in patients similarly to previous animal models [94] and could contribute to the susceptibility to neuroinflammation.

### 6.1. Genetic and Racial Factors

Genetic studies indicate that race may be a risk factor for SAE development. As such, the study group of Samuels and Girard was the first to investigate the role of mitochondrial DNA (mtDNA) in sepsis and SAE. In a DNA cohort extracted from the BioVU database and the Synthetic Derivative of Vanderbilt University Medical Center, the researchers demonstrated, after age adjustment, a relative risk of 1.36 (95% CI: 1.13–1.64; *p*-value = 0.001) in Caucasians belonging to mtDNA haplogroup IWX in comparison to other haplogroups among Caucasians and a relative risk of 0.60 (95% CI: 0.38–0.94; *p*-value = 0.03) in African Americans belonging to mtDNA haplogroup L2 regarding the development of SAE, as illustrated in Table 1 [95].

**Table 1 cells-12-01088-t001:** Non-modifiable risk factors of SAE.

Non-Modifiable Risk Factors		
Age		
Research article	Incidence of SAE/Distribution of SAE and non-SAE	Mortality of SAE patients
Zhang L et al. (2012) [96] Retrospective analysis	No statistically significant association Mean age 54y ± 18 SAE vs. 51y ± 14 non-SAE (*p*-value = 0.30)	n/a
Sonneville R et al. (2017) [37] Retrospective analysis	OR 1.02 (95% CI: 1.01–1.02; *p*-value < 0.01) Per 1-year increment	HR 1.03 (95% CI: 1.02–1.03; *p*-value < 0.01) Per 1-year increment
Feng Q et al. (2019) [97] Retrospective analysis	Mean age 55y ± 14 SAE vs. 57y ± 15 non-SAE (*p*-value = 0.334)	n/a
Chen J et al. (2020) [92] Retrospective analysis	n/a	OR 1.059 (95% CI: 1.027–1.093; *p*-value < 0.001) Per 1-year increment (28-day mortality)
Jin G et al. (2022) [89] Retrospective analysis	OR 1.054 (95% CI: 1.019–1.089; *p*-value = 0.002) Median age 77y (IQR 69–83.75) SAE vs. 73.5y (IQR 65–81) non-SAE (*p*-value = 0.012)	n/a
Lei W et al. (2022) [91] Prospective longitudinal study	SAE mean age 70.59 ± 8.56 vs. non-SAE mean age 56.36 ± 16.81 (*p*-value = 0.022)	n/a
Lu X et al. (2022) [90] Retrospective analysis	Increasing age as independent risk factor (cutoff 72y according to Shapley additive explanation/SHAP model) Mean age 68.1y ± 16.44 SAE vs. 66.33y ± 16.26 non-SAE (*p*-value < 0.001)	n/a
Genetic and racial factors		
Research article	Genetical factor	Incidence of SAE
Samuels DC et al. (2019) [95] Retrospective analysis	mtDNA haplogroup	Age-adjusted RR 1.36 (95% CI: 1.13–1.64; *p*-value = 0.001) Haplogroup clade IWX (7% caucasians) vs. Haplogroup H (most common in Caucasians) Age-adjusted RR 0.60 (95% CI: 0.38–0.94; *p*-value = 0.03) Haplogroup L2 (24% African Americans) vs. Haplogroup L3 (most common in African Americans)
Research article	Racial factor	Distribution of SAE and non-SAE
Lu X et al. (2022) [90] Retrospective analysis (*SAE n* = *4684, non-SAE n* = *4251*)	Ethnicity	In Asian cohort: incidence of SAE 2.4% vs. non-SAE 2.1% (*p*-value = 0.007; overall ethnic subgroups)

Similarly, Lu et al. (2022) reported a slightly higher incidence of SAE in Asian cohorts (2.4% SAE vs. 2.1% non-SAE; *p*-value = 0.007) compared to other ethnic cohorts (white: 69.8% SAE vs. 72.3% non-SAE; black: 8.9% SAE vs. 10.2% non-SAE; Hispanic: 2% SAE vs. 2.1% non-SAE; others: 17% SAE vs. 13.3% non-SAE) (*p*-value = 0.007), when analyzing the sepsis patients in the open-source Medical Information Mart for Intensive Care (MIMIC-VI) database (*n* = 8935) [90].

These findings suggest that some ethnicities are associated with more SAE development, while others seem to exhibit a protective factor.

Regarding HMGB1 levels, Chen et al. (2020) demonstrated a significantly increased level in a black ethnic cohort in comparison with a white ethnic cohort [94], which correlates with previous evidence of higher levels of pro-inflammatory markers such as hs-CRP and IL-6 in black ethnic groups [98,99,100]. Further research is warranted to determine potential racial protective or risk factors regarding the development of SAE.

### 6.2. Cardiovascular Diseases

Some studies investigated whether pre-existing cardiovascular disease (e.g., arterial hypertension, coronary artery disease (CAD), congestive heart failure (CHF), arrhythmias) significantly increase the risk of SAE. While Jin et al. (2022) found pre-existing CAD to be a risk factor for SAE development [85], other studies were not able to confirm this [91,96].

Similarly, discrepancies remain whether hypertension, by itself, can be considered to be a risk factor [37,89,90,91,96].

However, according to Sonneville et al. (2017) chronic cardiac disease is associated with a higher SAE mortality (HR 1.33, 95% CI: 1.10–1.61; *p*-value = 0.003) [37]. Neither arrhythmias [89] nor CHF [37,92] are, on the contrary, significantly associated with an increased risk of SAE. Studies detailing the effect of cardiovascular diseases on SAE development are tabulated (Table 2); the heterogeneity suggests that further research is needed to clarify their relevance as a risk factor.

In stress-induced hypertensive mice, HMGB1 was involved in microglia-mediated neuroinflammation, which in turn increased sympathetic vasoconstriction [101]. The question remains unanswered whether this holds true in human models. With respect to arrhythmias such as atrial fibrillation, HMGB1 has been identified to be a potential factor in the pathogenesis [102,103] and increased plasma levels have been found in patients suffering from CAD [104,105] which were even significantly associated with a higher mortality [106].

### 6.3. Respiratory Diseases

Similar to cardiovascular diseases, the evidence on respiratory diseases and their association with risk of SAE is inconsistent due to a lack of large studies showing conclusive evidence.

Jin et al. (2022) suggested that chronic obstructive pulmonary disease (COPD) in an elderly study population (age > 60 years; *n* = 222) was associated with a lower risk of SAE (16.4% SAE vs. 27.8% non-SAE; *p*-value = 0.047) and lower mortality in the SAE cohort (20% survival vs. 13.9% non-survival; *p*-value = 0.008) [89]. This finding is inconsistent with other results described in Table 2 [37,91,92,96].

**Table 2 cells-12-01088-t002:** Pre-existing conditions leading to SAE.

Pre-Existing Conditions		
Cardiovascular Disease		
Research article	Pre-existing condition	Distribution of SAE and non-SAE
Zhang L et al. (2012) [96] Retrospective analysis (*SAE n* = *41, non-SAE n* = *191*)	Coronary heart disease	9.8% SAE vs. 9.4% non-SAE (*p*-value = 0.573)
	Arterial hypertension	19.5% SAE vs. 19.4% non-SAE (*p*-value = 0.567)
Sonneville R et al. (2017) [37] Retrospective analysis (*SAE n* = *1341, non-SAE n* = *1172*)	Arterial hypertension	38.2% SAE vs. 35.4% non-SAE (*p*-value = 0.15)
Chen J et al. (2020) [92] Retrospective analysis (*SAE n* = *127, non-SAE n* = *164*)	Arterial hypertension	32.3% SAE vs. 20.7% non-SAE (*p*-value = 0.025)
Jin G et al. (2022) [89] Retrospective analysis (*SAE n* = *132, non-SAE n* = *90*)	Coronary heart disease	33.3% SAE vs. 18.9% non-SAE (*p*-value = 0.018)
	Arterial hypertension	57.6% SAE vs. 57.8% non-SAE (*p*-value = 0.976)
Lei W et al. (2022) [91] Prospective longitudinal study (*SAE n* = *17, non-SAE n* = *11*)	Coronary heart disease	29.4% SAE vs. 27.3% non-SAE (*p*-value = 0.624)
	Arterial hypertension	35.3% SAE vs. 54.5% non-SAE (*p*-value = 0.441)
Respiratory disease		
Research article	Pre-existing condition	Mortality of SAE
Jin G et al. (2022) [89] Retrospective analysis (*SAE n* = *132, non-SAE n* = *90*)	COPD	20% survival vs. 13.9% non-survival (*p*-value = 0.008)
Research article	Pre-existing condition	Distribution of SAE and non-SAE
Zhang L et al. (2012) [96] Retrospective analysis (*SAE n* = *41, non-SAE n* = *191*)	Obstructive lung disease	7.3% SAE vs. 5.8% non-SAE (*p*-value = 0.718)
Sonneville R et al. (2017) [37] Retrospective analysis (*SAE n* = *1341, non-SAE n* = *1172*)	COPD	10.9% SAE vs. 8.8% non-SAE (*p*-value = 0.08)
Chen J et al. (2020) [92] Retrospective analysis (*SAE n* = *127, non-SAE n* = *164*)	COPD	7.9% SAE vs. 11.0% non-SAE (*p*-value = 0.374)
Jin G et al. (2022) [89] Retrospective analysis (*SAE n* = *132, non-SAE n* = *90*)	COPD	16.7% SAE vs. 27.8% non-SAE (*p*-value = 0.047)
Lei W et al. (2022) [91] Prospective longitudinal study (*SAE n* = *17, non-SAE n* = *11*)	COPD	11.8% SAE vs. 9.1% non-SAE (*p*-value = 0.664)
Chronic liver disease		
Research article	Mortality from SAE	
Sonneville R et al. (2017) [37] Retrospective analysis (*SAE n* = *1341, non-SAE n* = *1172*)	HR 1.92 (95% CI: 1.54–2.39; *p*-value < 0.01) 9.5% SAE vs. 4.7% non-SAE (*p*-value < 0.01)	
Research article	Distribution of SAE and non-SAE	
Chen J et al. (2020) [92] Retrospective analysis (*SAE n* = *127, non-SAE n* = *164*)	3.1% SAE vs. 4.9% non-SAE (*p*-value = 0.462)	
Jin G et al. (2022) [89] Retrospective analysis (*SAE n* = *132, non-SAE n* = *90*)	4.5% SAE vs. 6.7% non-SAE (*p*-value = 0.553)	
Lei W et al. (2022) [91] Prospective longitudinal study (*SAE n* = *17, non-SAE n* = *11*)	29.4% SAE vs. 27.3% non-SAE (*p*-value = 0.624)	
Chronic alcohol abuse		
Research article	Incidence of SAE/Distribution of SAE and non-SAE	
Sonneville R et al. (2017) [37] Retrospective analysis (*SAE n* = *1341, non-SAE n* = *1172*)	OR 3.38 (95% CI: 2.34–4.89; *p*-value < 0.01) 11% SAE vs. 3.6% non-SAE (*p*-value < 0.01)	
Diabetes mellitus		
Research article	Distribution of SAE and non-SAE	
Zhang L et al. (2012) [96] Retrospective analysis (*SAE n* = *41, non-SAE n* = *191*)	7.3% SAE vs. 11% non-SAE (*p*-value = 0.471)	
Sonneville R et al. (2017) [37] Retrospective analysis (*SAE n* = *1341, non-SAE n* = *1172*)	19.7% SAE vs. 14.9% non-SAE (*p*-value < 0.01)	
Chen J et al. (2020) [92] Retrospective analysis (*SAE n* = *127, non-SAE n* = *164*)	18.1% SAE vs. 10.4% non-SAE (*p*-value = 0.057)	
Jin G et al. (2022) [89] Retrospective analysis (*SAE n* = *132, non-SAE n* = *90*)	28.8% SAE vs. 24.4% non-SAE (*p*-value = 0.474)	
Lei W et al. (2022) [91] Prospective longitudinal study (*SAE n* = *17, non-SAE n* = *11*)	35.3% SAE vs. 45.5% non-SAE (*p*-value = 0.701)	
Lu X et al. (2022) [90] Retrospective analysis (*SAE n* = *4684, non-SAE n* = *4251*)	26.3% SAE vs. 30.7% non-SAE (*p*-value = 0.001)	
Neurological and psychiatric conditions		
Research article	Pre-existing condition	Incidence of SAE
Sonneville R et al. (2017) [37] Retrospective analysis (*SAE n*=*1341, non-SAE n* = *1172*)	Neurological disease	OR 1.56 (95% CI: 1.18–2.06; *p*-value < 0.01) 13.6% SAE vs. 8.2% non-SAE (*p*-value < 0.01)
	Cognitive impairment	OR 2.25 (95% CI: 1.09–4.67; *p*-value = 0.03) 2.9% SAE vs. 0.9% non-SAE (*p*-value < 0.01)
	Chronic psychoactive drugs	OR 1.37 (95% CI: 1.11–1.70; *p*-value < 0.01) 22.1% SAE vs. 16% non-SAE (*p*-value *<* 0.01)
Research article	Pre-existing condition	Distribution of SAE and non-SAE
Sonneville R et al. (2017) [37] Retrospective analysis (*SAE n* = *1341, non-SAE n* = *1172*)	Stroke (neurological disease)	7.4% SAE vs. 3.9% non-SAE (*p*-value *<* 0.01)
	Epilepsy (neurological disease)	2.6% SAE vs. 1.0% non-SAE (*p*-value *<* 0.01)
Jin G et al. (2022) [89] Retrospective analysis (*SAE n* = *132, non-SAE n* = *90*)	Stroke	33.3.% SAE vs. 21.1% non-SAE (*p*-value = 0.047)
Chronic immunodepression		
Research article	Pre-existing condition	Mortality of SAE
Sonneville R et al. (2017) [37] Retrospective analysis (*SAE n* = *1341, non-SAE n* = *1172*)	Chronic immunodepression	HR 1.58 (95% CI: 1.34–1.87; *p*-value < 0.01) 25.2% SAE vs. 35.2% non-SAE (*p*-value < 0.01)
Research article	Pre-existing condition	Distribution of SAE and non-SAE
Sonneville R et al. (2017) [37] Retrospective analysis (*SAE n* = *1341, non-SAE n* = *1172*)	Chronic steroid use	6.3% SAE vs. 7.1% non-SAE (*p*-value = 0.46)
Chen J et al. (2020) [92] Retrospective analysis (*SAE n* = *127, non-SAE n* = *164*)	Immune diseases	1.8% SAE vs. 0.8% non-SAE (*p*-value = 0.803)

Although it remains unclear whether respiratory diseases increase the risk of SAE development, studies have demonstrated a correlation between severity of chronic rhinosinusitis with polyposis [107,108] and asthma [109,110,111] and the levels of HMGB1 in the sputum. In smoking patients, HMGB1 is significantly upregulated in the blood and lung tissues (epithelium, submucosal area and alveoli) in the presence of COPD, in contrast to smokers without COPD or healthy individuals with no expression of HMGB1 in the lung tissue [112]. Research indicates that HMGB1 levels in bronchoalveolar fluid and epithelial lining fluid correspond indirectly with spirometric values, i.e., forced expiratory volume during 1 s (FEV1), forced vital capacity (FVC) and FEV1/FVC of COPD patients [113,114], illustrating the degree of pulmonary inflammation and its detrimental effect on lung functioning.

### 6.4. Chronic Liver Disease

Some studies have pointed out that patients at risk for SAE have more chronic liver disease (CLD) and alcohol abuse history [37], although this remains inconsistent with other studies [89,91,92]. Of note, CLD was associated with a higher mortality rate (HR 1.92; 95% CI: 1.54–2.39; *p*-value < 0.01) in patients suffering from SAE [37] (Table 2).

In non-alcoholic fatty liver disease (NAFLD), significantly higher HMGB1 plasma levels have been detected in pediatric patients [115], confirming non-alcoholic steatohepatitis (NASH) animal models that demonstrated a high level HMGB1 and upregulation of RAGEs and TLRs as evidenced by increased mRNA levels in liver specimens [116,117]. HMGB1 seems to play a role in the pathogenesis of liver fibrosis in many diverse contexts, as evidenced by the significantly increased plasma levels compared with healthy control groups; e.g., hepatitis B virus (HBV)-induced CLD [118,119,120], carbon tetrachloride (CCl_4_) -induced liver fibrosis [121] and in the context of schistosomiasis [122]. Several studies have been undertaken to evaluate HMGB1 as a biomarker in plasma of CLD patients to monitor liver fibrosis [118,119]. Currently no clear answer is known, although it seems HMGB1 plasma levels are increased in early stages of liver injury and are more indicative of hepatocyte injury than fibrosis progression [118].

### 6.5. Diabetes Mellitus

Another debatable risk factor for SAE development is diabetes mellitus (DM). While one large cohort study (*n* = 2513) identified DM as a significant risk factor of SAE [37], in a larger cohort (*n* = 8935) DM seems to be significantly associated with less SAE development [90]. In DM patients, significantly higher plasma levels of HMGB1 have been found in comparison with healthy subjects [123]. Additionally, the increased pro-inflammatory state caused by the alarmins in an early phase of DM [124] is considered to be involved in the pathogenesis of complications of DM [125], such as endothelial dysfunction [126] and the subsequent CAD [104,127]. Considering the preceding evidence, the role of HMGB1 in this context with regard to development of SAE deserves further exploration.

### 6.6. Neurological and Psychiatric Conditions

Previous studies showed an association between pre-existing dementia and delirium in the ICU [128,129]. Table 2 demonstrates the association of increased SAE development with history of neuropsychiatric conditions, such as stroke [36,85], epilepsy, pre-existing cognitive impairment and chronic use of psychoactive medication (i.e., benzodiazepines or anti-psychotics) [90].

The role of HMGB1 is being investigated in the context of possible novel treatments targeting the DAMP in several neurological conditions, such as stroke, subarachnoid hemorrhage, traumatic brain injury, epilepsy and neurodegenerative diseases [82]. HMGB1 is thought to be involved in the ictogenesis of epileptic insults [130], and anti-HMGB1 treatment seems to be beneficial for forms of epilepsy in several animal models [131]. In Alzheimer’s disease, evidence demonstrates the involvement of HMGB1 in the aggregation of beta-amyloid [75]. Several studies have shown the role of HMGB1 in Parkinson’s disease with respect to dopaminergic neuronal cell death in animal models and, in this context, anti-HMGB1 treatments show a positive response and demonstrate the protective effect on neuronal cell death and blood–brain barrier disruption [132,133].

### 6.7. Chronic Immunodepression

Sonneville et al. (2017) showed that fewer chronically immunodepressed patients tend to develop SAE, though these patients have higher mortality rates associated with SAE [36].

While a relatively small study suggested that patients with previous immune diseases are less likely to develop SAE [(92)], in patients suffering from rheumatoid arthritis (RA), an autoimmune disease characterized by a higher state of inflammation, extracellular HMGB1 is known to be responsible for stimulating pro-inflammatory cytokine release (namely TNF and IL-1) [134].

Additionally, in animal models comparing rheumatoid arthritis with osteoarthritis significantly higher levels of HMGB1 were found in synovial fluid in the context of RA [135]. Direct recombinant HMGB1 joint injection triggered a pro-inflammatory reaction by activating macrophages and inducing production of IL-1 via NF-κB activation [136].

To the best of our knowledge, currently no research has explored the direct link between HMGB1, chronic immunodepression and SAE.

## 7. Characteristics of SAE Patients at Admission to the Intensive Care Unit (ICU)

### 7.1. Disease Severity Scores

In 1985, Knaus et al. revised the Acute Physiology and Chronic Health Evaluation score (APACHE score), renamed APACHE-II, which equipped clinicians with a tool to assess and classify the severity of acute illness and proved to be prognostically reliable as well [137]. In the following decade, the group of Vincent published a sepsis-specific score on behalf of the European Society of Intensive Care Medicine (ESICM): Sepsis-related Organ Failure Assessment (SOFA) score [138]. Both scores are widely used in the assessment of sepsis [32].

Regarding SAE, higher APACHE-II scores at ICU admission are significantly associated with a higher risk of development of SAE (OR 1.239; 95% CI: 1.144–1.341; *p*-value < 0.001) [92] and higher APACHE-II scores are observed in SAE groups in contrast to non-SAE groups [89,91,92,96,97] and are associated with higher SAE mortality (OR 1.178; 95% CI: 1.063–1.305; *p*-value < 0.01) [92]. Similarly, higher SOFA scores at ICU admission are consistently associated with higher risk of development of SAE [89,90,92,97]. Sonneville et al. (2017) observed several organ-specific SOFA elements (excluding the Glasgow Coma Scale (GCS)) to be significantly associated with higher risk and mortality of SAE [37]. The relation between higher disease severity scores and increased SAE development have been highlighted in Table 3 for each scoring system.

The association between higher SOFA scores and mortality of SAE is consistent with other studies [89,92]. 

Interestingly, several studies have reported significant correlations between ICU plasma HMGB1 levels of septic patients and SOFA scores [139,140]. Plasma HMGB1 levels also increased from day 1 to 3 post-ICU admission in non-survivor septic patients (*p* < 0.02). Day 3 post-ICU admission plasma HMGB1 levels were also indicative of mortality in septic patients (area under the ROC curve 0.71, 95% confidence interval 0.51–0.91, *p* = 0.05) [140].

### 7.2. Site of Infection and Identified Pathogen

Several studies found significant associations with higher risk of SAE and the initial site or type of infection: biliary tract [96], gastrointestinal tract [92,96], pulmonary infections and endocarditis [37].

Some infection sites were associated with a significantly lower risk of SAE according to Sonneville et al. (2017): intra-abdominal, skin and soft tissue and catheter-related infections [37]. Smaller studies could not reproduce all these findings, possibly due to the sample size [89,92,96].

Large studies have found a potential association between the identified pathogen and the development of SAE [37,90], although the question remains if some pathogens are more associated with SAE than others.

Based on several studies, it seems certain pathogens are more frequently identified in SAE patients: *Acinetobacter* [96], *Pseudomonas aeruginosa* [96], *Stenotrophomonas maltophila* [96], *Staphylococcus aureus* [37,96], *Enterococcus faecium* [92,96] and *Escherichia coli* [97]. However, inconsistency among the studies does not answer the question whether a specific pathogen is more likely to be a risk factor of SAE than another [37,89,92,97].

Colonization by specific pathogens at time of ICU admission is associated with increased risk of infection and in some cases (i.e., Enterococcus or vancomycin-resistant Enterococcus/VRE) even with increased mortality [141].

As the etiology regarding SAE development is multifactorial, SAE might be associated with specific pathogen-related toxins [142,143] and potential neuroinflammation [144] caused by disturbed microbiota due to previous antibiotic treatments [145,146]. A summary of the findings, focusing on the site of infection and each specific pathogen in the development of SAE, is tabulated (Table 3).

Of note, median plasma HMGB1 levels in patients with B. pseudomallei sepsis were significantly higher than in non-B. pseudomallei sepsis patients (11.1 ng/mL vs. 7.1 ng/mL, *p* = 0.001). Among those with B. pseudomallei sepsis, plasma HMGB1 levels were higher in those with a 1-month outcome of death than in those who survived (median 14.8 ng/mL vs. 9.2 ng/mL, *p* = 0.038). Whether these higher plasma HMGB1 levels were associated with SAE development was, however, not investigated in this study [139].

## 8. ICU Management of SAE

The mainstay of SAE management is based on the early detection of delirium (often the first manifestation of sepsis itself), determination of the underlying cause, accurate and prompt treatment of the infection and provision of supportive care.

### 8.1. SAE Diagnosis

SAE remains a diagnosis of exclusion, which can only be made after other infective and non-infective causes of encephalopathy have been ruled out in sepsis. Hepatic, uremic or respiratory encephalopathy, metabolic disturbances, drug overdose, withdrawal of sedatives or opioids, alcohol withdrawal delirium and Wernicke’s encephalopathy are the main differential diagnoses of sepsis-associated encephalopathy [147].

Diagnosis of SAE requires a multimodal approach, including clinical, neurological and cognitive examinations. Clinical examination of patients with SAE can reveal a level of consciousness (regardless of administered sedative treatment) and might show disturbances of sleep–wake cycles or evidence of hallucinations, restlessness or agitation, among other symptoms commonly seen in delirium. Apart from the abnormal mental status and paratonic rigidity, the findings in the neurological examination are usually unremarkable with cranial nerve function invariably spared [33]. Therefore, a series of complementary examinations should be carried out to finalize the diagnosis of SAE. 

### 8.2. EEG

Electroencephalography (EEG) is used to exclude not only convulsive and non-convulsive seizures but also sepsis-induced periodic discharges [34]. Non-convulsive and sepsis-induced periodic discharges have been reported in 11% and 25%, respectively, of 98 patients with sepsis-related alterations in mental status hospitalized in a medical ICU [148].

The EEG of SAE patients is characterized by a diffuse cortical dysfunction with generalized EEG slowing and the presence of theta and delta waves [149]. Moreover, the lack of EEG reactivity is the only consistent predictor of unfavorable outcomes (such as ICU mortality) in SAE patients [34,148,150,151].

### 8.3. Transcranial Doppler (TCD)

TCD is easy to perform at point of care to estimate cerebral blood flow velocities that are altered in most patients suffering from SAE [152]. Shramm et al. observed (through TCD correlated with systemic arterial blood pressure) an impairment of cerebral autoregulation in 83% of 30 septic patients, especially during the two first days of sepsis. Seventy-six percent of these patients with altered cerebral autoregulation developed signs of SAE [153].

## 9. Pharmacological Management of SAE

Currently, there are no specific guidelines on the pharmacological management of sepsis-associated encephalopathy. 

Recent randomized clinical trials have not evidenced any benefit of statins and dexmedetomidine for the prevention or treatment of delirium in septic patients, despite their respective anti-neuroinflammatory and noradrenergic/anti-apoptotic properties [154].

Enhanced sensitivity towards benzodiazepines is present in systemic inflammatory processes and, in general, lorazepam, a highly potent benzodiazepine, should be avoided.

Recent studies have pointed towards the potential benefits of dexmedetomidine. Pandharipande et al. compared the effects of dexmedetomidine to those of lorazepam sedation in septic ICU patients. Dexmedetomidine-sedated septic patients had 3.2 more delirium/coma-free days on average (95% CI for difference, 1.1 to 4.9), 1.5 (−0.1, 2.8) more delirium-free days and 6 (0.3, 11.1) more ventilator-free days (*p* < 0.05) [155].

Several studies have also pointed out a decrease in 28/90-day mortality rates when dexmedetomidine is used [155,156].

Interestingly, in vitro data have directly indicated that dexmedetomidine decreases LPS-upregulated HMGB1 levels in the cytoplasm (*p* < 0.05) [157].

Taken together, despite these benefits, it does remain difficult to use dexmedetomidine as a sole sedation agent (particularly for deep sedation, Richmond Agitation Sedation Scale (RASS) of −3 to −5), often requiring propofol/midazolam association [158].

Although the routine use of anti-psychotic agents in the treatment of delirium is discouraged, patients who experience significant distress secondary to symptoms of delirium such as anxiety, fearfulness, hallucinations or delusions, or who are agitated and may be physically harmful to themselves or others, may benefit from short-term use of haloperidol or an atypical anti-psychotic until these distressing symptoms resolve [159].

## 10. Non-Pharmacological Management of SAE

Non-pharmacological interventions should be promoted and should be applied in all septic patients. These include early mobilization, discontinuation of sedation, rehydration, management of physical and psychological discomfort, improving cognition and optimizing sleep, vision and hearing and time and space orientation [159].

## 11. Unknowns

Sepsis-associated encephalopathy is thought to be the most common cause of encephalopathy in the medical–surgical ICU.

Due to SAE’s complex pathophysiology, it remains a challenge for intensivists.

An unambiguous definition, as well as a clear diagnostic tool to better identify patients, is firstly very much needed.

Risk factors for SAE development include age, genetic factors, as well as pre-existing neuropsychiatric conditions. Sepsis due to certain infection sites/origins might be more prone to encephalopathy development than others.

Currently ICU management of SAE is mainly based on non-pharmacological support.

Although there are limited data available about the role of HMGB1 in neuroinflammation following sepsis, it has been implicated in other neurologic disorders, where its translocation from the nucleus to the extracellular space has been found to trigger neuroinflammatory reactions (such as regulating M1/M2 phenotypic polarization of microglia) and damage the blood–brain barrier.

HMGB1 could therefore be a target in mitigating SAE development.

## Figures and Tables

**Figure 1 cells-12-01088-f001:**
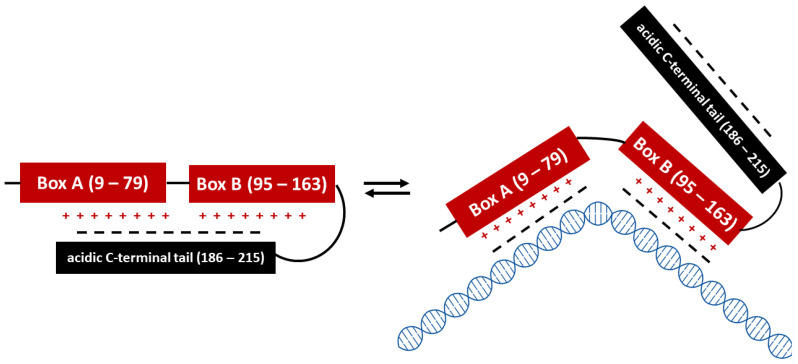
Dynamic equilibrium of unbound and DNA-bound HMGB1 structure. Schematic representation of human HMGB1. The three domains are represented by colored boxes. Red: positively charged DNA-binding Boxes A and B, black: negatively charged acidic tail. Interactions between positively charged Boxes A and B and the negatively charged acidic tail regulate HMGB1 DNA-binding activities by competitive electrostatic interactions.

**Figure 2 cells-12-01088-f002:**
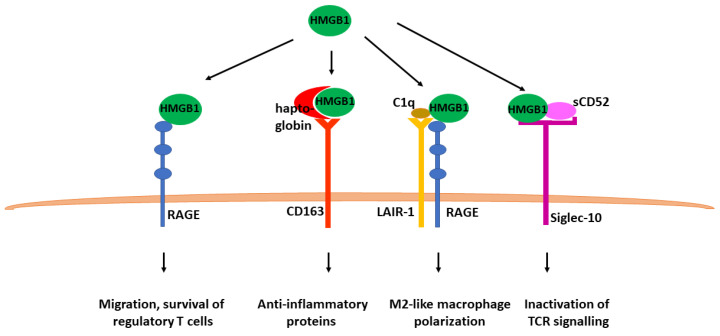
Anti-inflammatory activities of HMGB1. HMGB1 mediates several anti-inflammatory activities through binding to several receptors alone or in association with soluble mediators.

**Figure 3 cells-12-01088-f003:**
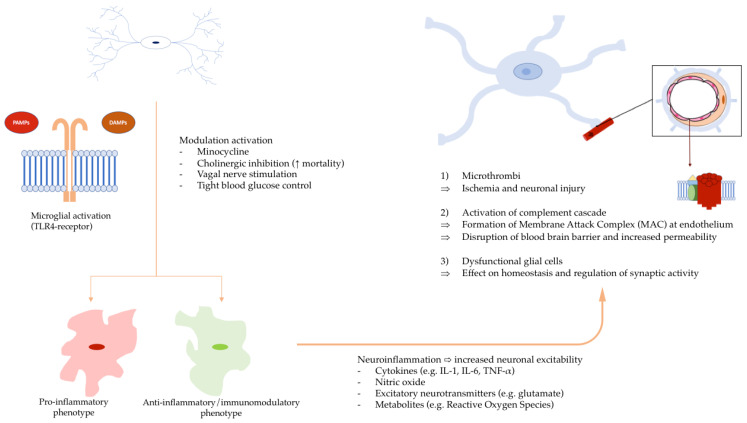
Microglial activation leading to neuroinflammation. Microglia can take on either anti- or pro-inflammatory phenotypes. In their pro-inflammatory state, cytokines, nitric oxide, excitatory neurotransmitters and metabolites (such as reactive oxygen species) are released in the brain parenchyma. Multiple experimental interventions have shown promising results in modulating microglial activity. In addition to the direct effects of microglial activation on neurons, neuroinflammation may also contribute to SAE through its effects on the microvasculature of the brain. Inflammation can lead to the formation of microthrombi in the brain, which can cause ischemia and further neuronal injury. Additionally, the activation of the complement cascade can lead to the formation of the membrane attack complex (MAC), which can damage the endothelial cells, leading to blood–brain barrier breakdown and increased permeability.

**Table 3 cells-12-01088-t003:** Patient characteristics at ICU admission and incidence of SAE.

Patient Characteristics at ICU Admission			
Severity scores			
Research article	Score	Incidence of SAE/Distribution of SAE and non-SAE	Mortality of SAE
Zhang L et al. (2012) [96] Retrospective analysis (*SAE n* = *41, non-SAE n* = *191*)	APACHE-II	Mean score 22 ± 7 SAE vs. 17 ± 7 non-SAE (*p*-value = 0.000)	n/a
Sonneville R et al. (2017) [37] Retrospective analysis (*SAE n* = *1341, non-SAE n* = *1172*)	Modified SOFA (non-neurological SOFA)	Median score 7 (IQR 5–10) SAE vs. 6 (IQR 3–8) non-SAE (*p*-value *<* 0.01)	HR 1.10 (95% CI: 1.08–1.12; *p*-value < 0.01) Per 1-point increment
	Renal SOFA >2	OR 1.41 (95% CI: 1.19–1.67; *p*-value *<* 0.01) 62.6% SAE vs. 50.4% non-SAE (*p*-value *<* 0.01)	n/a
	Respiration SOFA >2	69.7% SAE vs. 58.3% non-SAE (*p*-value *<* 0.01)	n/a
	Liver SOFA >2	28.5% SAE vs. 23.4% non-SAE (*p*-value *<* 0.01)	n/a
Feng Q et al. (2019) [97] Retrospective analysis (*SAE n* = *74, non-SAE n* = *101*)	APACHE-II	Mean score 19 ± 6 SAE vs. 13 ± 6 non-SAE (*p*-value *<* 0.001)	n/a
	SOFA	Mean score 11 ± 4 SAE vs. 8 ± 5 non-SAE (*p*-value *<* 0.001)	n/a
Chen J et al. (2020) [92] Retrospective analysis (*SAE n* = *127, non-SAE n* = *164*)	APACHE-II	OR 1.239 (95% CI: 1.144–1.341; *p*-value *<* 0.001) Median score 15 (IQR 12–19.5) SAE vs. 9 (IQR 6–11) non-SAE (*p*-value *<* 0.01)	OR 1.178 (95% CI: 1.063–1.305; *p*-value *<* 0.01) Per 1-point increment (28-day mortality)
	SOFA	OR 1.421 (95% CI: 1.244–1.623; *p*-value *<* 0.001) Median score 7 (IQR 5–10) SAE vs. 3 (IQR 1–5) non-SAE (*p*-value *<* 0.01)	OR 1.167 (95% CI: 1.009–1.349; *p*-value = 0.037) Per 1-point increment (28-day mortality)
Jin G et al. (2022) [89] Retrospective analysis (*SAE n* = *132, non-SAE n* = *90*)	SOFA	Median score 10 (IQR 6–13) SAE vs. 4 (IQR 2–7) non-SAE (*p*-value *<* 0.001)	OR 1.185 (95% CI: 1.074–1.307; *p*-value < 0.001)
	APACHE-II	Median score 23 (IQR 15–13) SAE vs. 12 (IQR 9–17) non-SAE (*p*-value *<* 0.001)	n/a
Lei W et al. (2022) [91] Prospective longitudinal study (*SAE n* = *17, non-SAE n* = *11*)	APACHE-II	Median score 24 (IQR 18–28) SAE vs. 15 (IQR 12–20) non-SAE (*p*-value = 0.005)	n/a
Lu X et al. (2022) [90] Retrospective analysis (*SAE n* = *4684, non-SAE n* = *4251*)	SOFA	Mean score 5.58 ± 2.71 SAE vs. 4.27 ± 2.15 non-SAE (*p*-value *<* 0.001)	n/a
Site of infection and identified pathogen			
Research article	Site/type of infection	Incidence of SAE/Distribution of SAE and non-SAE	
Zhang L et al. (2012) [96] Retrospective analysis (*SAE n* = *41, non-SAE n* = *191*)	Biliary tract	24.39% SAE vs. 12.57 non-SAE (*p*-value = 0.05)	
	Intestines	43.90% SAE vs. 27.23% non-SAE (*p*-value = 0.029)	
	Pulmonary	41.46% SAE vs. 33.51% non-SAE (*p*-value = 0.214)	
	Skin and soft tissue	19.51% SAE vs. 11.52% non-SAE (*p*-value = 0.131)	
Sonneville R et al. (2017) [37] Retrospective analysis (*SAE n* = *1341, non-SAE n* = *1172*)	Intra-abdominal	23.4% SAE vs. 27.8% non-SAE (*p*-value = 0.01)	
	Pulmonary	32% SAE vs. 24.6% non-SAE (*p*-value *<* 0.01)	
	Endocarditis	2.2% SAE vs. 1% non-SAE (*p*-value = 0.02)	
	Skin and soft tissue	OR 0.57 (95% CI: 0.39–0.82; *p*-value < 0.01) 4.4% SAE vs. 7.3% non-SAE (*p*-value < 0.01)	
	Catheter-related	OR 0.53 (95% CI: 0.32–0.88; *p*-value = 0.01) 2.1% SAE vs. 4.4% non-SAE (*p*-value < 0.01)	
Chen J et al. (2020) [92] Retrospective analysis (*SAE n* = *127, non-SAE n* = *164*)	Biliary tract	6.3% SAE vs. 10.4% non-SAE (*p*-value = 0.22)	
	Gastrointestinal tract	26.8% SAE vs. 15.2% non-SAE (*p*-value = 0.015)	
	Pulmonary	59.8% SAE vs. 50.6% non-SAE (*p*-value = 0.076)	
	Skin and soft tissue	3.9% SAE vs. 6.1% non-SAE (*p*-value = *0.408*)	
Jin G et al. (2022) [89] Retrospective analysis (*SAE n* = *132, non-SAE n* = *90*)	Biliary tract	9.1% SAE vs. 13.3% non-SAE (*p*-value = 0.318)	
	Gastrointestinal tract	7.6% SAE vs. 2.2% non-SAE (*p*-value = 0.129)	
	Intra-abdominal	13.6% SAE vs. 17.8% non-SAE (*p*-value = 0.40)	
	Pulmonary	78.8% SAE vs. 65.6% non-SAE (*p*-value = 0.028)	
	Skin and soft tissue	3.0% SAE vs. 6.7% non-SAE (*p*-value = 0.20)	
Research article	Identified pathogen	Incidence of SAE/Distribution of SAE and non-SAE	
Zhang L et al. (2012) [96] Retrospective analysis (*SAE n* = *41, non-SAE n* = *191*)	*Acinetobacter*	39.02% SAE vs. 17.28% non-SAE (*p*-value = 0.005)	
	*P. aeruginosa*	24.39% SAE vs. 8.38% non-SAE (*p*-value = 0.011)	
	*S. maltophila*	12.19% SAE vs. 1.05% non-SAE (*p*-value = 0.002)	
	*S. aureus*	12.19% SAE vs. 3.14% non-SAE (*p*-value = 0.028)	
	*E. faecium*	24.39% SAE vs. 9.42% non-SAE (*p*-value = 0.015)	
	*E. coli*	26.83% SAE vs. 14.14% non-SAE (*p*-value = 0.061)	
Sonneville R et al. (2017) [37] Retrospective analysis (*SAE n* = *1341, non-SAE n* = *1172*)	*P. aeruginosa*	7.2% SAE vs. 6.1% non-SAE (*p*-value = 0.24)	
	*S. aureus*	OR 1.54 (95% CI: 1.05–2.25; *p*-value = 0.03) 6.9% SAE vs. 4.4% non-SAE (*p*-value *<* 0.01)	
	*Enterobacteriaceae*	28.3% SAE vs. 30.4 non-SAE (*p*-value = 0.26)	
Feng Q et al. (2019) [97] Retrospective analysis (*SAE n* = *74, non-SAE n* = *101*)	*Acinetobacter*	21.6% SAE vs. 12.87% non-SAE (*p*-value = 0.124)	
	*P. aeruginosa*	12.16% SAE vs. 3.96% non-SAE (*p*-value = 0.083)	
	*S. aureus*	9.46% SAE vs. 7.92% non-SAE (*p*-value = 0.719)	
	*E. coli*	35.14% SAE vs. 20.79% non-SAE (*p*-value = 0.031)	
Chen J et al. (2020) [92] Retrospective analysis (*SAE n* = *127, non-SAE n* = *164*)	*Acinetobacter*	21.3% SAE vs. 20.7% non-SAE (*p*-value = 0.913)	
	*Pseudomonas*	6.3% SAE vs. 6.1% non-SAE (*p*-value = 0.944)	
	*Staphylococcus*	12.6% SAE vs. 11.6% non-SAE (*p*-value = 0.792)	
	*Enterococcus*	16.5% SAE vs. 7.9% non-SAE (*p*-value = 0.023)	
	*E. coli*	24.4% SAE vs. 21.9% non-SAE (*p*-value = 0.621)	
Jin G et al. (2022) [89] Retrospective analysis (*SAE n* = *132, non-SAE n* = *90*)	*A. baumannii*	30.3% SAE vs. 25.6% non-SAE (*p*-value = 0.441)	
	*P. aeruginosa*	10.6% SAE vs. 8.9% non-SAE (*p*-value = 0.674)	
	*S. maltophila*	6.82% SAE vs. 5.6% non-SAE (*p*-value = 0.704)	
	*Staphylococcus*	16.7% SAE vs. 14.4% non-SAE (*p*-value 0.656)	
	*Enterococcus*	6.82% SAE vs. 8.9% non-SAE (*p*-value = 0.569)	
	*E. coli*	10.6% SAE vs. 12.2% non-SAE (*p*-value = 0.708)	

## Data Availability

Not applicable.

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
