# Peer review of "High Mobility Group Box 1 (HMGB1): Potential Target in Sepsis-Associated Encephalopathy"

_cells, 2023, doi:10.3390/cells12071088_

Round 1

Reviewer 1 Report

In this review, the authors provide an overview of septic encephalopathy, the role and function of HMGB1 in its pathogenesis and the potential of novel therapeutic agents targeting it. The overall focus of the article is well organized and easy for the reader to understand. On the other hand, there were a few points that could have been added to the figures and text content to make the review better, and we would like to suggest these.

Major points:

1. With regard to Figure 1, the structure of HMGB1 is not considered to be so important, as it is well described in other reviews. On the other hand, the authors demonstrated the binding region between HMGB1 and DNA in the manuscript, and we wonder if that information could be added to the figure.

2. Figure 2 is also covered in other reviews. Rather, the mechanism by which HMGB1 is neutralized via several mechanisms, introduced on page 3, line 117, is newer, more useful information and more appropriate as a figure.

3. In Section 5, it has been reported in the past that metformin binds directly to HMGB1 and inhibits its inflammatory effects (Horiuchi et al., J. Biol. Chem., 2017). This information should also be added.

4. In Section 6.2, this review should also add information on autoimmune diseases, in particular rheumatoid arthritis, where HMGB1 and inflammation are known to be involved.

Minor points:

1. Cited papers should be added for some sentences. Page 3, line 80: “~ its redox state.”, Page 3, line 82: “~HMGB1 isoforms.”, Page 3, line 86: “~CXCL12 alone.”., Page 3, line 90: “~monocytes/macrophages.”

2. Other sentences state MD2-TLR4, but only on Page 3, line 91, TLR4-MD2.

3. In some places, there appeared to be unnecessary space. Page 1, line 39, Page 3, line 82.

4. On page 4, line 144, there were two "with" followed by.

Author Response

Point to point reply to the reviewers’ comments

Thank you for your comments. We have addressed each point and have highlighted changes in the manuscript.

Upon your suggestion, table 1 and figure 4 have been removed.

Reviewer 1

In this review, the authors provide an overview of septic encephalopathy, the role and function of HMGB1 in its pathogenesis and the potential of novel therapeutic agents targeting it. The overall focus of the article is well organized and easy for the reader to understand. On the other hand, there were a few points that could have been added to the figures and text content to make the review better, and we would like to suggest these.

Major points:

  1. With regard to Figure 1, the structure of HMGB1 is not considered to be so important, as it is well described in other reviews. On the other hand, the authors demonstrated the binding region between HMGB1 and DNA in the manuscript, and we wonder if that information could be added to the figure. 

This has been changed.

  1. Figure 2 is also covered in other reviews. Rather, the mechanism by which HMGB1 is neutralized via several mechanisms, introduced on page 3, line 117, is newer, more useful information and more appropriate as a figure.

This has been changed.

  1. In Section 5, it has been reported in the past that metformin binds directly to HMGB1 and inhibits its inflammatory effects (Horiuchi et al., J. Biol. Chem., 2017). This information should also be added. This has been changed.

  1. In Section 6.2, this review should also add information on autoimmune diseases, in particular rheumatoid arthritis, where HMGB1 and inflammation are known to be involved.

This has been added.

Minor points:

  1. Cited papers should be added for some sentences. Page 3, line 80: “~ its redox state.”, Page 3, line 82: “~HMGB1 isoforms.”, Page 3, line 86: “~CXCL12 alone.”., Page 3, line 90: “~monocytes/macrophages.”

This has been changed.

  1. Other sentences state MD2-TLR4, but only on Page 3, line 91, TLR4-MD2.

This has been changed.

  1. In some places, there appeared to be unnecessary space. Page 1, line 39, Page 3, line 82.

This has been changed.

  1. On page 4, line 144, there were two "with" followed by.

This has been changed.

Reviewer 2 Report

The authors summarize the current understandings of HMGB1 regulations, actions, and pathological roles as well as sepsis-associated encephalopathy (SAE) pathology and discuss the causal link between HMGB1 and SAE pathology and its clinical implications based on updated basic and clinical studies. Given the clinical significance of SAE and the lack of its mechanistic understanding, this manuscript is informative for the broad readership of the journal Cells. However, the manuscript is not well organized in its current form, primarily because some sections solely describe the characteristics of SAE without relating to HMGB1, distracting the whole manuscript from its main theme, the role of HMGB1 in SAE as a potential therapeutic target. Therefore, the authors should revise the manuscript before publication by clarifying their points and removing or at least shortening the unnecessary sections to discuss the role of HMGB1 in SAE. Specific comments are provided below.

Major comments

1.     The abstract should include the conclusions of this manuscript.

2.     It is widely accepted that microglia can take many more states. Thus, microglial experts claim to abandon this dualistic classification (PMID 36327895). Since many studies still mention the M1 and M2 states of microglia, the authors might be unable to avoid using these terminologies. Yet the authors should still warn the readers to prevent their usage.

3.     Table 1 should be cited and explained in appropriate places.

4.     Several descriptions of clinical studies include the percentage of patient populations. The authors should clarify what the denominators are in each instance.

5.     Sections 7.1, 8, 9, 11, and 12 are irrelevant to HMGB1. The authors should remove or shrink these sections unless they cannot mention their relevance to HMGB1.

6.     Table 4 should be cited and explained in appropriate places.

7.     This reviewer does not find Figure 4 helpful and suggests the authors remove it.

Minor comments

1.     There are several grammatical errors. The authors should consult a native speaker for English proofreading.

2.     Line 32: “HMGBA” should be “HMGA.”

3.     Line 36: “comprise” should be “compose.”

4.     Lines 49-51: Figure 2 citation should help.

5.     Line 54: “karyopherin α1” needs a brief explanation.

6.     Line 66: “beclin” needs a brief explanation.

7.     Lines 137-139: These sentences are misleading and should be “Disulfide HMGB1 (dsHMGB1) signals via TLR4 and RAGE. Sulfonyl HMGB1 (oxHMGB1) only binds to RAGE.”

8.     Line 144: “with” is duplicated.

9.     Line 147: A space is needed between “to” and “>.”

10.  Line 155: Is “15/” correct?

11.  Line 167: “structures” is ambiguous. Should it be “brain structures”?

12.  Line 180: See major comment 2.

13.  Line 211: Figure 3: See major comment 2.

14.  Line 260: See major comment 2.

15.  Line 261: The first “in” is unnecessary.

16.  Lines 280-281: More information on the antagonists should be provided.

17.  Lines 288-289: Where these changes were seen in the body should be mentioned.

18.  Lines 327-331: See major comment 4.

19.  Lines 341-359: Table 3 should be cited and explained in section 6.2.1.

20.  Lines 360-378: Table 3 should be cited and explained in section 6.2.2.

21.  Lines 365-367: See major comment 4.

22.  Line 376: “FEV1 and FEV1/FVC” needs a brief explanation.

23.  Line 386-387: Where these changes were seen in the body should be mentioned.

24.  Lines 426-436: Table 3 should be cited and explained in section 6.2.4.

25.  Lines 429: The meaning of the following sentence is unclear: “this effect is significantly less observed in SAE patients.”

26.  Lines 437-451: Table 3 should be cited and explained in section 6.2.5.

27.  Lines 452-456: Table 3 should be cited and explained in section 6.2.6.

28.  Line 564: “the key elements” should be removed?

29.  Lines 564-566: The meaning of the following sentence is unclear: “Clinical examination of patients with SAE can reveal a level of consciousness not corresponding to the degree of administered sedative treatment.”

30.  Line 569: “is” should be removed?

31.  Line 583: “TCD is an easy to perform bedside exam” should be “TCD is easy to perform in a bedside exam”?

32.  Line 624: Figure 4: See major comment 2.

33.  Lines 636: See major comment 2.

Author Response

Reviewer 2

The authors summarize the current understandings of HMGB1 regulations, actions, and pathological roles as well as sepsis-associated encephalopathy (SAE) pathology and discuss the causal link between HMGB1 and SAE pathology and its clinical implications based on updated basic and clinical studies. Given the clinical significance of SAE and the lack of its mechanistic understanding, this manuscript is informative for the broad readership of the journal Cells. However, the manuscript is not well organized in its current form, primarily because some sections solely describe the characteristics of SAE without relating to HMGB1, distracting the whole manuscript from its main theme, the role of HMGB1 in SAE as a potential therapeutic target. Therefore, the authors should revise the manuscript before publication by clarifying their points and removing or at least shortening the unnecessary sections to discuss the role of HMGB1 in SAE. Specific comments are provided below.

Major comments

  1. The abstract should include the conclusions of this manuscript. This has been changed.
  2. It is widely accepted that microglia can take many more states. Thus, microglial experts claim to abandon this dualistic classification (PMID 36327895). Since many studies still mention the M1 and M2 states of microglia, the authors might be unable to avoid using these terminologies. Yet the authors should still warn the readers to prevent their usage.

We fully agree with this comment. However, given the current lack of experimental studies using the new microglial nomenclature approach, we continue to use the "simplistic" M1/M2 description in our manuscript. Nevertheless, we have modified the manuscript to warn readers about the new international recommendations regarding microglial nomenclature.

  1. Table 1 should be cited and explained in appropriate places. Table 1 has been removed.
  2. Several descriptions of clinical studies include the percentage of patient populations. The authors should clarify what the denominators are in each instance.

Denominators have been added in-text where appropriate and in the tables under each article respectively, which should provide readers with an adequate estimation of the study populations.

  1. Sections 7.1, 8, 9, 11, and 12 are irrelevant to HMGB1. The authors should remove or shrink these sections unless they cannot mention their relevance to HMGB1. This has been changed.
  2. Table 4 should be cited and explained in appropriate places. This has been changed.
  3. This reviewer does not find Figure 4 helpful and suggests the authors remove it. 

 Figure 4 has been removed.

Minor comments

  1. There are several grammatical errors. The authors should consult a native speaker for English proofreading.

This has been done.

  1. Line 32: “HMGBA” should be “HMGA.” This has been changed.
  2. Line 36: “comprise” should be “compose.” This has been changed.
  3. Lines 49-51: Figure 2 citation should help. This has been added.
  4. Line 54: “karyopherin α1” needs a brief explanation.

This has been changed.

  1. Line 66: “beclin” needs a brief explanation.

This has been changed.

  1. Lines 137-139: These sentences are misleading and should be “Disulfide HMGB1 (dsHMGB1) signals via TLR4 and RAGE. Sulfonyl HMGB1 (oxHMGB1) only binds to RAGE.” This has been changed.
  2. Line 144: “with” is duplicated. This has been changed.
  3. Line 147: A space is needed between “to” and “>.” This has been changed.
  4. Line 155: Is “15/” correct? This has been changed.
  5. Line 167: “structures” is ambiguous. Should it be “brain structures”? This has been changed to ‘neurological structures’.
  6. Line 180: See major comment 2. This has been changed.
  7. Line 211: Figure 3: See major comment 2. This has been changed.
  8. Line 260: See major comment 2. This has been changed.
  9. Line 261: The first “in” is unnecessary. This has been changed.
  10. Lines 280-281: More information on the antagonists should be provided. This has been added.
  11. Lines 288-289: Where these changes were seen in the body should be mentioned. This has been addressed.
  12. Lines 327-331: See major comment 4. This has been added.
  13. Lines 341-359: Table 3 should be cited and explained in section 6.2.1. This has been added.
  14. Lines 360-378: Table 3 should be cited and explained in section 6.2.2. This has been added.
  15. Lines 365-367: See major comment 4. This has been added.
  16. Line 376: “FEV1 and FEV1/FVC” needs a brief explanation. This has been addressed.
  17. Line 386-387: Where these changes were seen in the body should be mentioned. This has been addressed.
  18. Lines 426-436: Table 3 should be cited and explained in section 6.2.4. This has been added.
  19. Lines 429: The meaning of the following sentence is unclear: “this effect is significantly less observed in SAE patients.” This has been changed.
  20. Lines 437-451: Table 3 should be cited and explained in section 6.2.5. This has been added.
  21. Lines 452-456: Table 3 should be cited and explained in section 6.2.6. This has been added.
  22. Line 564: “the key elements” should be removed? This has been removed.
  23. Lines 564-566: The meaning of the following sentence is unclear: “Clinical examination of patients with SAE can reveal a level of consciousness not corresponding to the degree of administered sedative treatment.” This has been changed.
  24. Line 569: “is” should be removed? This has been changed.
  25. Line 583: “TCD is an easy to perform bedside exam” should be “TCD is easy to perform in a bedside exam”? This has been changed.
  26. Line 624: Figure 4: See major comment 2.

Figure 4 has been removed.

  1. Lines 636: See major comment 2. This has been changed.

Reviewer 3 Report

The authors put together the relationship between HMGB1 and sepsis-associated encephalopathy. I think that this manuscript is insufficient for the publication, and request to consider the following comments.

The figures are poor resolution. I cannot understand what it say especially in figure 3 and 4. The authors should correct these figures.

Table 1 and 4 are not shown in this manuscript.

I wonder whether Section 8-11 is necessary in this review.

Author Response

Reviewer 3

The authors put together the relationship between HMGB1 and sepsis-associated encephalopathy. I think that this manuscript is insufficient for the publication, and request to consider the following comments.

The figures are poor resolution. I cannot understand what it say especially in figure 3 and 4. The authors should correct these figures.

Figure 4 has been removed. Resolution of figure 3 has been updated.

Table 1 and 4 are not shown in this manuscript.

Table 1 has been removed. References to table 4 (table 3 now) have been added.

I wonder whether Section 8-11 is necessary in this review. This has been changed.

Round 2

Reviewer 2 Report

The authors have appropriately addressed the concerns raised by this reviewer. There is one typo, as below, but the authors can correct it when checking a galley proof.

Line 189: “M-M2” should be “M1-M2.”